# Systematic review of the effectiveness of selected drugs for preventive chemotherapy for *Taenia solium* taeniasis

**Michelle M. Haby**[1,2]*, **Leopoldo A. Sosa Leon**[3], **Ana Luciañez**[4], **Ruben Santiago Nicholls**[4], **Ludovic Reveiz**[5], **Meritxell Donadeu**[6,7]

1 Department of Chemical and Biological Sciences, University of Sonora, Hermosillo, Sonora, Mexico, 2 Centre for Health Policy, Melbourne School of Population and Global Health, The University of Melbourne, Melbourne, Victoria, Australia, 3 Independent consultant, Hermosillo, Sonora, Mexico, 4 Neglected Infectious Diseases, Communicable Diseases and Environmental Determinants of Health, Pan American Health Organization/World Health Organization, Washington DC, United States of America, 5 Department of Evidence and Intelligence for Action in Health, Pan American Health Organization/World Health Organization, Washington DC, United States of America, 6 Faculty of Veterinary and Agricultural Sciences, The University of Melbourne, Melbourne, Victoria, Australia, 7 Initiative for Neglected Animal Diseases (INAND), Midrand, South Africa

* haby@unimelb.edu.au

## Abstract

### Background

Preventive chemotherapy is a useful tool for the control of *Taenia solium* taeniasis and cysticercosis. The aim of this systematic review is to assess the scientific evidence concerning the effectiveness and safety of different drugs in preventive chemotherapy for *T. solium* taeniasis in endemic populations.

### Methods

A systematic review was conducted of controlled and uncontrolled studies, assessing the efficacy and adverse effects (among other outcomes) of albendazole, niclosamide and/or praziquantel for preventive chemotherapy of *T. solium* taeniasis. A comprehensive search was conducted for published and unpublished studies. Two reviewers screened articles, completed the data extraction and assessment of risk of bias. A meta-analysis of cure rate and relative reduction in prevalence was performed. The protocol for this review was registered on the International prospective register of systematic reviews (PROSPERO), number CRD42018112533.

### Results

We identified 3555 records, of which we included 20 primary studies reported across 33 articles. Meta-analyses of drug and dose showed that a single dose of praziquantel 10mg/kg, albendazole 400mg per day for three consecutive days, or niclosamide 2g, resulted in better cure rates for *T. solium* taeniasis (99.5%, 96.4% and 84.3%, respectively) than praziquantel 5mg/kg or single dose albendazole 400mg (89.0% and 52.0%, respectively). These findings

**Data Availability Statement:** All relevant data are within the manuscript and its Supporting Information files.

**Funding:** This review was funded by PAHO/WHO. The funders had no role in the study design, data collection or analysis, decision to publish, or preparation of the manuscript.

**Competing interests:** The authors have declared that no competing interests exist.

have a low certainty of evidence due to high risk of bias in individual studies and heterogeneity in combined estimates. In relation to side-effects, most studies reported either no or only mild and transient side-effects within the first three days following drug administration for all drugs and doses.

## Conclusion

Evidence indicated that praziquantel 10mg/kg, niclosamide 2g, and triple dose albendazole 400mg were effective as taenicides and could be considered for use in mass drug administration programs for the control of *T. solium* taeniasis. Evidence was not found that any of these drugs caused severe side effects at the indicated doses, although the extent of the available evidence was limited.

### Author summary

Taeniasis and cysticercosis caused by *Taenia solium* are considered, by the WHO, to be neglected tropical diseases. Preventive chemotherapy for taeniasis in endemic populations is a useful tool for control of the parasite. Preventive chemotherapy can be implemented by treating the whole population in endemic areas at regular intervals (known as mass drug administration). While different drugs, doses and regimes have been used there were still uncertainties about which drugs and dose have the best efficacy while considering adverse effects. We conducted a systematic review of the best available literature to inform a WHO guideline for preventive chemotherapy. We found that praziquantel 10mg/kg, niclosamide 2g, and triple dose albendazole 400mg (400mg per day for three consecutive days) are all effective. Proven side-effects were mild and short-lived, when they did occur.

## Introduction

*Taenia solium (T. solium)* is a parasite found almost exclusively in humans and pigs. It causes both taeniasis (in its adult form) and cysticercosis (in its metacestode larval form). The highest prevalence of *T. solium* is found in Latin America, sub-Saharan Africa, and east, south and south-east Asia [1, 2]. Endemic (and suspected endemic) countries for *T. solium* have been defined by WHO and shown on their updated map for 2015 [2].

Taeniasis is an intestinal infection caused by 3 species of tapeworm: *T. solium* (pork tapeworm), *Taenia saginata* (beef tapeworm) and *Taenia asiatica* ('Asian' tapeworm) [3]. Infection with *T. solium* tapeworm occurs when humans eat raw or undercooked, infected pork that contains mature, viable cysticerci (the larval stage of the tapeworm). Taeniasis due to *T. solium (as well as T. saginata* and *T. asiatica*) is usually characterized by mild and non-specific symptoms [3]. Abdominal pain, nausea, diarrhea or constipation may arise when the tapeworms, become fully developed in the intestine, approximately 8 weeks after ingestion of meat containing cysticerci (larvae).

Pigs develop the cysticercus lifecycle stage by ingesting tapeworm eggs released in the feces of a human infected with a *T. solium* tapeworm [4]. Cysticerci can develop in the muscles, eyes and the central nervous system of pigs. The tapeworm eggs are also infective for humans if they are ingested. In humans the parasite develops into a cysticercus larval stage similar to the metacestode which develops in pigs. The cysticerci commonly encyst in muscle tissues and the

central nervous system. When cysts develop in the brain and/or spinal cord, the condition is referred to as neurocysticercosis. Cerebral cysts are a major cause of adult onset seizures in most low-income countries [3]. The most frequent symptoms of neurocysticercosis include seizures and chronic headaches, although other manifestations also occur [3, 5].

Taeniasis/cysticercosis caused by *T. solium* is listed by the WHO as a neglected tropical disease. In 2011 the Neglected Tropical Diseases Advisory Group of the World Health Organization (WHO) developed a roadmap [6] for the control of the 17 neglected tropical diseases, which included *T. solium*. This roadmap was ratified by the Member States at the 66th World Health Assembly in Resolution WHA66.12 [7].

Various tools are available for the control and elimination of *T. solium*, one of which is preventive chemotherapy (PC) for taeniasis in humans [8]. It can be implemented in three ways: i) mass drug administration (MDA) when the whole population of a predefined geographical area is treated irrespective of clinical status; ii) targeted chemotherapy when only specific risk groups are treated; and iii) selective chemotherapy which screens patients and subsequently treats according to clinical status [9].

Some commercial anthelmintics have shown efficacy in the treatment of taeniasis, including albendazole (ALB), praziquantel (PZQ) and niclosamide (NICL). This review concerns the use of these three commercial products. NICL shows good efficacy as a single dose (2g for adults, and adjusted for children), and has little systemic absorption and therefore no adverse effect in people with neurocysticercosis but is usually more expensive than PZQ [10, 11]. PZQ has also been shown to be effective and is unrestricted for use in pregnant and lactating women [12]. It is routinely used in MDA programs targeting schistosomiasis. It has the drawback, however, of being systemically absorbed and can cross the blood-brain barrier, thus may have the potential to result in adverse neurological consequences stemming from the inflammatory response invoked by cysts that are damaged by the drug in patients with neurocysticercosis [10, 11, 13]. Triple dose ALB (3 x 400mg/person, given over three consecutive days) has also demonstrated good efficacy against *Taenia* spp. [14] but can also cross the blood-brain barrier. Single dose ALB 400mg is recommended for use in MDA programs targeting soil-transmitted helminths [15].

The objective of this review was to assess the scientific evidence for the use of preventive chemotherapy for *T. solium* taeniasis in endemic populations using NICL, PZQ or ALB. While the focus is their use in MDA, literature concerning treatment of taeniasis in any circumstance was also included to allow a better understanding of the effectiveness and adverse effects of different doses and regimens (frequency of application).

The broad review questions addressed by the review were: What is the effectiveness of PZQ, NICL, and ALB for preventive chemotherapy for the control of *T. solium* taeniasis in endemic populations? (*T. solium* endemic populations were defined as the populations in which the full cycle of the parasite transmission is present). What are the adverse effects associated with each drug? Should any population groups be excluded or monitored more closely due to potential adverse effects? The specific questions developed as PICO (population, intervention, comparison, outcomes) questions are shown in Box 1.

## Methods

High quality systematic review methods were used [16]. The protocol was registered on the International prospective register of systematic reviews (PROSPERO: CRD42018112533) [17] and the Preferred Reporting Items for Systematic Reviews and Meta-Analysis statement [18] (S1 Checklist) and the WHO handbook for guideline development [19] for reporting were followed.

Box 1. Specific PICO (population, intervention, comparison, outcomes) questions addressed by the systematic review.

1. In individuals with confirmed or suspected taeniasis due to *T. solium*, what are the doses to be used and the regimen for the treatment of *T. solium* taeniasis with NICL, PZQ or ALB?

    P: individuals with confirmed or suspected *T. solium* taeniasis

    I: antiparasitic treatment with NICL, PZQ or ALB at a particular dose and frequency

    C: the different treatments at lower doses or different frequency, or no medication

    O: lower infection rate with *T. solium* taenia (taeniasis by *T. solium*), e.g. prevalence, incidence, or cure rate.

    Examples: PZQ at 10mg/kg vs 5mg/kg as a single dose; repeated dose of NICL vs single dose of NICL.

2. In individuals living in *T. solium* endemic areas, do the potential side effects of NICL, PZQ or ALB at any dose and frequency justify their exclusion from PC for the control of taeniasis?

    P: individuals living in *T. solium* endemic areas

    I: antiparasitic treatment with NICL, PZQ or ALB at a particular dose and frequency

    C: the different treatments at lower doses or different frequency; treatment with NICL, PZQ or ALB for other parasites (e.g. schistosomiasis) in areas non-endemic for *T. solium*; or no treatment

    O: risk of side effects (seizures, severe headaches), lower infection rate with *T. solium* taeniasis

3. For each drug, should any population groups be excluded or monitored more closely due to potential adverse effects, e.g. pregnant (or suspected pregnant) women?

    P: pregnant (or suspected pregnant) women living in endemic areas to *T. solium*; individuals with symptoms consistent with cysticercosis; other at-risk groups

    I: antiparasitic treatment (NICL, PZQ or ALB)

    C: no medication or one of the above-mentioned drugs

    O: risk of side effects, lower infection rate with *T. solium* taeniasis

4. In school aged children in areas co-endemic with *T. solium* and soil-transmitted helminths, could PC for both parasites be given simultaneously?

    P: school aged children in areas co-endemic to *T. solium* and soil-transmitted helminths

    I: PC with NICL or PZQ for the treatment of taeniasis simultaneously with PC with ALB (used at a single dose of 400mg/person); ALB used at 400mg/person for 3 consecutive days

C: NICL or PZQ alone; single dose of ALB.

O: lower infection rate with *T. solium* taeniasis and soil-transmitted helminths, risk of side effects due to simultaneous medication

5. In individuals living in a *T. solium* endemic area that have been treated with PC, for how long should adverse effects be monitored, considering the potential side effects of the different mentioned drugs?

P: individuals living in a *T. solium* endemic area that have been treated with PC

I: PC with NICL, PZQ or ALB

C: no medication

O: observation time of side effects due to NICL, ALB or PZQ

## Inclusion criteria

**Participants.** Individuals (humans) of any age with confirmed or suspected *T. solium* taeniasis for PICO questions 1 and 3, or those living in *T. solium* endemic (or suspected endemic) areas (as defined by WHO [2]) for PICO questions 2, 4 and 5, including: pregnant (or suspected pregnant) women, school aged children in areas co-endemic to *T. solium* and soil-transmitted helminths, and individuals with asymptomatic neurocysticercosis.

**Interventions.** All forms of PC with NICL, PZQ or ALB were included in the review, i.e. mass drug administration, targeted and selective chemotherapy (including treatment). PC combined with other prevention and control measures (e.g. health education, vaccination of pigs) were included in measurement of efficacy only if the effect could be attributed to the PC. Other prevention and control measures tested in isolation from PC were excluded, e.g. health education, improved sanitation, vaccination and treatment of pigs. Treatment of cysticercosis or neurocysticercosis with NICL, PZQ or ALB was excluded.

**Comparisons.** No medication; alternative doses and regimens for PC; treatment with NICL, PZQ or ALB for other parasites (e.g. schistosomiasis) in areas non-endemic for *T. solium*.

**Outcomes.** Primary outcome measures included were: lower infection rate with *T. solium* taeniasis (e.g. prevalence, incidence, or cure rate); lower infection rate with soil-transmitted helminths (for PICO question 4); higher risk of side effects (e.g. seizures, severe headaches); higher risk of side effects due to simultaneous medication; observation time of side effects due to NICL, ALB or PZQ; costs, cost-effectiveness; feasibility; values and preferences of participants; impact on equity. One secondary outcome measure was included: porcine cysticercosis rate.

When measuring the effectiveness of an intervention in reducing the infection rate with *T. solium* taeniasis several diagnostic strategies have been used such as microscopy of feces to identify the eggs of the parasite, macroscopy to search for taeniid material in feces, detection of coproantigens, copro-DNA, detection of antibodies in serum, or combinations of these techniques [20]. The techniques with the best reported sensitivity and specificity include species-specific copro-DNA (PCR) and copro-Ag-ELISA [21]. Alternatively, measures that aren't

species-specific could be used to screen (e.g. fecal microscopy, serology, non-species-specific coproantigen), followed by confirmation of positive samples with species-specific techniques. Studies were not excluded based on the diagnostic test used.

**Study types.** The following study types were considered: systematic reviews, randomized controlled trials, non-randomized controlled trials, controlled before-after studies, interrupted-time-series studies with before and after measures, before-after studies, repeated measures studies, and economic evaluations (cost-benefit, cost-effectiveness, cost-utility). Qualitative studies were only included if they provided information on the values and preferences of participants for different MDA strategies or on their feasibility. Modelling studies were excluded for efficacy/effectiveness but not for economic evaluations. Case reports were excluded because they are a very low level of evidence of effect and chance cannot be ruled out.

Studies published in English, French, Spanish or Portuguese were included, with no date of publication limitations.

### Search strategy

The following databases were searched from inception to date of search: CAB Abstracts, PubMed, EMBASE, LILACS, SciELO, and Cochrane central register of controlled trials (CENTRAL). Specialized sources of systematic reviews and economic evaluations searched included: Cochrane Database of Systematic Reviews, Database of Abstracts of Reviews of Effects, Epistemonikos, Health Technology Assessments, and NHS Economic Evaluation Database. Some of these databases index a combination of published and unpublished studies (for example, doctoral dissertations and conference abstracts) therefore unpublished studies were partially captured through the electronic search process. Supplementary sources searched using the same keywords included: International Initiative for Impact Evaluation (3ie), Google and Google Scholar, System for Information on Grey Literature in Europe (Open grey– www.opengrey. eu); WHO International Clinical Trials Registry Platform, reference list of included studies; reference list of systematic reviews; reference list of key WHO/PAHO documents [e.g. 8, 22]. Experts on the topic were also asked to identify additional studies.

The search terms included MeSH terms (where relevant for the database) and key words in the title, abstract and/or as key words. The PICO (Participants, Intervention, Comparison and Outcomes) framework and published studies on the topic of *Taenia solium* were used to identify relevant search terms. In practice, the search strategy mainly included text words and MeSH terms related to the condition (*Taenia solium*, Taenia) and the intervention (albendazole, niclosamide, praziquantel, chemoprevention, mass drug administration, anthelmintics, deworm) to ensure a sensitive search. Variations of the words in Spanish, French and Portuguese were included–where the search platform supports foreign characters.

Searches were conducted by one review author and references imported into Endnote. The date of last search of electronic databases was 26 September 2018. Duplicates were removed before screening. The search strategy and results for each of the databases can be found in S1 File.

### Study selection

The screening of the titles and abstracts against the inclusion criteria was conducted by two review authors (MMH and LASL) independently and the full text of any potentially relevant papers identified by either reviewer was retrieved for closer examination. The inclusion criteria were applied independently against these papers by two reviewers. Disagreements regarding eligibility of studies were resolved via discussion and consensus. A third reviewer (MD) was consulted where any doubts remained. All studies that initially appeared to meet the inclusion

criteria but on inspection of the full text paper did not were detailed in a table, together with reasons for their exclusion.

## Data extraction

Two reviewers (MMH and LASL) independently extracted all relevant data from the included papers. Differences in interpretation by the two reviewers were resolved by discussion and consensus. Data extracted included: country and year of study, study design, sample size calculation reported; details of participants: N, age group, gender, socioeconomic status, and specific characteristics (e.g. pregnant women); intervention: type of PC, drug, dose, number of doses and time between treatments, other strategies where applicable, population coverage achieved (for community-based studies); follow-up period; comparisons; outcomes measured; results; possible conflicts of interest; and comments on research gaps etc.

## Risk of bias and quality of the evidence assessment

The risk of bias of each of the included studies was assessed independently by two reviewers (MMH and LASL). Primary studies were assessed using the Cochrane Effective Practice and Organization of Care (EPOC) Review Group tool, which is an adaptation of the Cochrane Collaboration "Risk of Bias" tool–with some minor modifications [16, 23, 24] (S2 File). The advantage of this tool is that it is suitable for various study designs and is commonly used in both Cochrane and non-Cochrane systematic reviews.

The Grading of Recommendations Assessment, Development and Evaluation (GRADE) (www.gradeworkinggroup.org) approach to grading quality (or certainty) of evidence and strength of recommendations was used to assess the body of evidence for each PICO question [25, 26].

## Strategy for data synthesis

A narrative synthesis and meta-analysis of the main outcome variable (infection rate with *T. solium* taeniasis) was conducted. To measure the infection rate with *T. solium* taeniasis the majority of studies utilized either the cure rate expressed as a percentage, or the relative reduction in prevalence from baseline to follow-up. Cure rate is an indicator of the efficacy of a drug, while the relative reduction in prevalence is an indicator of the effectiveness of the drug in MDA and will be influenced by other factors such as population coverage, time of follow-up and sampling. Both are influenced by the diagnostic method. A meta-analysis was undertaken using the inverse variance heterogeneity method, which is an improved alternative to the random effects method, and using MetaXL version 5.3 (Ersatz, EpiGear International, Sunrise Beach, Australia) [27, 28]. The double arcsine transformation of prevalence was used [29]. Heterogeneity was assessed using Cochran's Q and $I^2$ statistics. Doi plots and the Luis Furuya–Kanamori (LFK) index were used to evaluate the presence of small-study effects, where asymmetry can indicate publication or other biases [27, 30, 31]. A narrative synthesis of the evidence was also undertaken, and summary tables produced.

## Analysis of subgroups or subsets

A number of subgroup (and/or sensitivity) analyses were planned, including population subgroup, study quality, method of determining *T. solium* taeniasis burden, and type/severity of side-effect; however, an insufficient number of studies was available. Subgroup

analyses were conducted of drug and dose for the main outcome measure of lower infection rate with *T. solium* taeniasis, measured as cure rate or relative reduction in prevalence.

## Results

We identified 3555 records after removal of duplicates. We excluded 3299 records based on the screening of titles and abstract against the inclusion criteria and assessed the full-text of the remaining 256 records for eligibility. The selection process for studies and the numbers at each stage are shown in Fig 1. In all, 35 articles met the inclusion criteria of which 2 were systematic reviews [32, 33]–these were used as a source of studies to prevent double-counting of studies. We included 20 primary studies–reported across 33 articles–in the systematic review and excluded 221 (S3 File). For the 33 articles we identified 20 as the primary reference [34–53] and 13 as supporting references [13, 14, 54–64]. A source for the full text of some studies (n = 90) could not be found for assessment; 78% of these were published before 1990. Two studies with no full text available appear to meet the inclusion criteria [65, 66] and another four are preliminary reports (conference abstracts) of studies that appear to meet the inclusion criteria but not yet published in full [67–70]. Language was the principal reason for exclusion for 30 studies, two of which appeared to otherwise meet the inclusion criteria [71, 72]. On

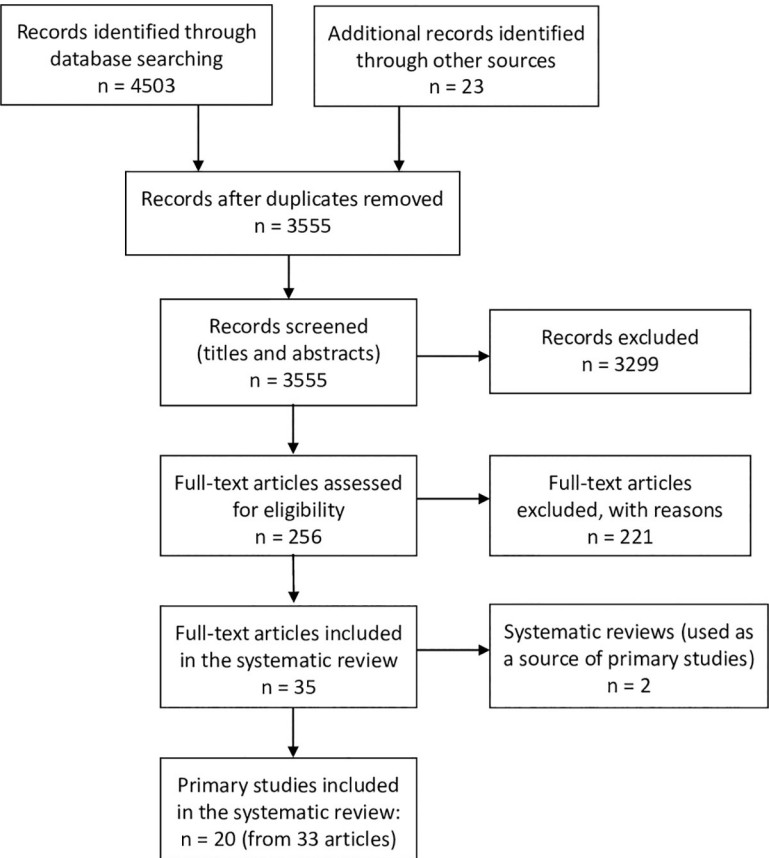

**Fig 1. Study selection flow diagram–Preventive chemotherapy for the control of taeniasis by *Taenia solium* in endemic areas.**

further examination, one of these appears to be reporting the same results as an included study [49].

## Characteristics of included studies

The characteristics of the included studies are summarized in Tables 1 and 2 and details of the *T. solium* taeniasis diagnostic test strategies are shown in S1 Table. The majority of studies were conducted in Central and South America (n = 10), followed by Asia (n = 8) and Africa (n = 2). Eleven of the studies used a before-after study design, while nine used a controlled design: randomized controlled trial (n = 2), controlled before-after study (n = 6) or controlled trial (n = 1). Most of the studies included participants of all ages, except for one of MDA in school-aged children [35], one of selective chemotherapy in children aged 4–6 years [52] and another only in adults [44].

## Intervention

The drug tested was ALB in 7 studies, NICL in 4 studies, PZQ in 7 studies. A combination of PZQ and NICL was used in 2 studies: one used 40mg/kg PZQ for the MDA in school-aged children (done as part of the National Schistosomiasis Control Programme) and NICL at 50mg/kg for children (2g for adults) or PZQ 10mg/kg for selective chemotherapy [35]; and the other used 2g NICL for selective chemotherapy for taeniasis, ALB for other intestinal parasites and then 10mg/kg PZQ for the MDA [39]. Thus, for these two studies [35, 39], it was not possible to attribute the effect (lower infection rate with *T. solium* taeniasis) to one drug.

Ten of the studies tested MDA as an intervention (Table 1), of which three tested it as the only intervention [48, 49, 51]. Three of these 10 studies also included selective chemotherapy of cases [34, 35, 39] and three tested MDA plus education [37, 42, 50]–with one of the intervention arms in the study by Steinmann et al. also including construction of latrines [50]. One of the MDA interventions included education as well as vaccination and preventive chemotherapy in pigs [46]–thus the efficacy results cannot be attributed to the MDA but the outcomes related to side-effects, feasibility, and values and preferences can be used.

Ten of the studies tested selective chemotherapy as the main intervention (Table 2), of which two also included education [43, 45] and one first screened pigs for cysticercosis, then screened residents living within a 100-meter ring surrounding any heavily-infected pig [45].

## Risk of bias in included studies

All of the included studies were classified as having a high risk of bias because they scored 'high risk' or 'unclear risk' on more than two of the assessment criteria [73]. Of the nine studies that had a control group, only two were randomized controlled trials, which is the best study design for measuring efficacy. The most common limitations were: allocation concealment (n = 7 studies scored high risk), random sequence generation (n = 7 high risk), baseline outcomes similar (n = 4 high risk), incomplete outcome data addressed (n = 4 high risk), blinding of participants and personnel (n = 2 high risk), and blinding of outcome assessors (n = 8 unclear risk) (Fig 2 and S1 Fig).

Of the 11 studies that used a before-after design the worst scoring item was 'bias in selection of participants into the study' (n = 8 high risk)–with most studies either using a very select group of participants (for selective chemotherapy) or unable to show that they had a random sample of the population (for baseline measures before MDA), e.g. due to low response rate and no data presented on non-responders (Fig 2 and S1 Fig). The next worst scoring item was

**Table 1. Characteristics of included studies that tested Mass Drug Administration–with or without selective chemotherapy (n = 10).**

| Study ID and references | Country and year/s of study | Study type | Participants: number, age, specific characteristics | Intervention | Drug and dose | Pop. coverage (%)[a] | Follow-up period | Outcomes |
|---|---|---|---|---|---|---|---|---|
| Allan et al. 1997 [34] | Guatemala 1994–1996 | BA | n = 2019; all ages. | Selective chemotherapy of cases; MDA of uninfected individuals (2–3 months later) | NICL 2g ($\geq$ 6 years), 1g (<6 years) | 74.9% | 10 months | • Infection rate with TS taeniasis–prevalence • Porcine cysticercosis rate —seroprevalence |
| Braae et al. 2017 [35, 55–57, 60] | Tanzania 2012–2015 | CBA | n = 951 Mbozi; school aged children (4–15 years); area co-endemic for schistosomiasis[b] n = 880 Mbeya | IntA—Annual MDA (3 times over 2 years) plus selective chemotherapy[c] (3 times over 2½ years) plus information IntB—Biennial MDA (2 times over 2 years) plus selective chemotherapy[c] (3 times over 2½ years) plus information | PZQ 40mg/kg for MDA; NICL 50mg/kg or PZQ 10mg/kg for selective chemotherapy | NR | 9–10 months (post final round of MDA) 6–7 months (post final track-and-treat) | • Infection rate with TS taeniasis–prevalence • Risk of side-effects • Porcine cysticercosis rate —seroprevalence |
| | | | n = 561 Mbozi, n = 621 Mbeya; adults > 15 years; area co-endemic for schistosomiasis[b] | IntC—Selective chemotherapy[c] (3 times over 2½ years) plus information. | NICL 2g or PZQ 10mg/kg—3 times over 3 years | NR | 6–7 months (post final track-and-treat) | |
| Cruz et al. 1989 [37, 58] | Ecuador 1985–1987 | BA | n = 739 at follow-up; all ages $\geq$ 6 years; person with a history of epilepsy, allergies, pregnant women, or those severely ill excluded. | MDA plus education | PZQ 5mg/kg | 75.8% in houses examined | 1 year | • Infection rate with TS–prevalence • Risk of side-effects • Costs • Feasibility • Values and preferences of participants • Impact on equity • Porcine cysticercosis rate —prevalence |
| Diaz Camacho 1991 [39] | Mexico 1988–1989 | BA | n = 559; all ages > 5 years; pregnant women and persons with liver cirrhosis or neurologic symptoms excluded. | Selective chemotherapy of cases and then MDA | 2g NICL for selective chemotherapy of taeniasis[d]; PZQ 10mg/kg for MDA | 60.6% (71% of eligible pop.) | 1-year | • Infection rate with TS taeniasis–prevalence • Infection rate with STH–prevalence • Porcine cysticercosis rate —prevalence |
| Keilbach 1989 [42] | Mexico 1986–1987 | BA | n = 760; all ages $\geq$ 5 years; persons suspected to suffer from neurocysticercosis did not receive PZQ but NICL instead. | MDA plus education | PZQ 5mg/kg | $\approx$60% | 4 months | • Feasibility • Values and preferences of participants • Porcine cysticercosis rate —prevalence & seroprevalence |

(*Continued*)

**Table 1.** (Continued)

| Study ID and references | Country and year/s of study | Study type | Participants: number, age, specific characteristics | Intervention | Drug and dose | Pop. coverage (%)[a] | Follow-up period | Outcomes |
|---|---|---|---|---|---|---|---|---|
| Okello 2016 [14, 46, 54] | Lao People's Democratic Republic 2013–2015 | BA | n = 298; all ages > 6 years; persons with acute illness, pregnant and lactating women excluded. | MDA—twice 6 months apart—plus education plus pigs vaccinated with TSOL18 and treated with oxfendazole at 30mg/kg —three times 6 months apart[e]. | ALB 400mg per day for three consecutive days | MDA1– 64% MDA2– 63% (>85% of eligible pop.) | 10 months (post MDA2) | • Infection rate with TS taeniasis[e]–prevalence<br>• Infection rate with STH–prevalence<br>• Risk of side-effects<br>• Feasibility<br>• Values and preferences of participants |
| Sarti 2000 [13, 48] | Mexico 1991–1996 | BA | n = 3007; all ages > 4 years; persons with hepatic disease and pregnant women excluded. | MDA | PZQ 5mg/kg | 87% | 6 and 42 months | • Infection rate with TS taeniasis–prevalence<br>• Risk of side-effects<br>• Porcine cysticercosis rate —prevalence & seroprevalence |
| Steinmann 2008 [51] | China 2007 | RCT | n = 66 ALB, n = 57 Tribendimidine; all ages (range 5–87 years); persons with acute illness, pregnant, or who had drunk alcohol on the day of treatment excluded. Area co-endemic for soil-transmitted helminths. | Int—MDA with ALB Cont—MDA with Tribendimidine | ALB 400mg ($\geq$ 15 years), 200mg (5 to 14 years); Tribendimidine 400mg ($\geq$ 15 years), 200mg (5 to 14 years) | NA | 2–4 weeks | • Infection rate with TS taeniasis–cure rate & prevalence<br>• Infection rate with STH–cure rate & prevalence<br>• Risk of side-effects |
| Steinmann 2011 [49, 64] | China 2008 | RCT | n = 314; all ages $\geq$ 5 years; persons with chronic disease or other conditions likely to interfere with treatment, pregnant women, recent anthelminthic treatment excluded. Area co-endemic for soil-transmitted helminths. | MDA: IntA—single dose ALB IntB—triple dose ALB given over 3 consecutive days ContA—single dose mebendazole ContB—triple dose mebendazole given over 3 consecutive days | ALB 400mg Mebendazole 500mg | NA | 3–5 weeks | • Infection rate with TS taeniasis —cure rate & prevalence<br>• Infection rate with STH—cure rate & prevalence<br>• Risk of side-effects<br>• Values and preferences of participants<br>• Observation time of side-effects |

(*Continued*)

**Table 1.** (Continued)

| Study ID and references | Country and year/s of study | Study type | Participants: number, age, specific characteristics | Intervention | Drug and dose | Pop. coverage (%)[a] | Follow-up period | Outcomes |
|---|---|---|---|---|---|---|---|---|
| Steinmann 2015 [50] | China 2007–2013 | CBA | n = 760, 100 samples per village; all ages ≥ 2years; persons with acute or chronic illness, pregnant women excluded. Area co-endemic for soil-transmitted helminths. | IntA—Annual MDA; IntB—6-monthly MDA; IntC—6-monthly MDA + latrine construction + regular health education[f] (3-year period). Then annual MDA by local village doctors for another 2-year period. | ALB 400mg | 80–90% of the eligible pop. (phase one of study) | ≈2 years and 5 years following baseline measure | • Infection rate with TS taeniasis[f]–prevalence • Infection rate with STH—prevalence |

Abbreviations: ALB—albendazole; BA—before-after study; CBA—controlled before-after study; Cont—control; CT—controlled trial; Int—intervention; MDA—mass drug administration; NICL—niclosamide; NR—not reported; pop.—population; PZQ—praziquantel; RCT—randomized controlled trial; STH—soil-transmitted helminths; TS—*Taenia solium*

[a] coverage of total population unless otherwise stated

[b] MDA of children done as part of National Schistosomiasis Control Programme.

[c] small number of pigs also treated with oxfendazole 30mg/kg

[d] selective chemotherapy for other intestinal parasites also given: ALB 400mg per day for three consecutive days for *Hymenolepis nana*; ALB 400mg for *ascaris*, *enterobius*, and *Trichuris*; and metronidazole, 20mg per day for five days for *Giardia lamblia* and *Entamoeba histolytica*.

[e] efficacy results cannot be attributed to MDA given the intervention in pigs. Only information on side-effects can be used.

[f] efficacy results for intervention C cannot be attributed to MDA given the construction of latrines as well as more intense education.

'incomplete outcome data addressed' (n = 7 high risk)–with most studies of MDA having an even lower response rate at follow-up and no data presented on non-responders. None of the studies explicitly stated that they used blinding of outcome assessors.

For all findings, the GRADE quality assessment gave a low or very low level of certainty. These are specified when referring to each of the outcomes.

### Effects by outcome

**Infection rate with *T. solium* taeniasis.**  Nineteen of the primary studies measured the infection rate with *T. solium* taeniasis but one of these could not be used to measure efficacy of preventive chemotherapy because there was also an intervention in pigs [46]. Of the 18 studies used for this outcome, eight measured the reduction in prevalence between baseline and follow-up following MDA as the principal intervention [34, 35, 37, 39, 48–51] (Table 1), of which two also measured the cure rate [49, 51]. Another nine measured the cure rate following selective chemotherapy as the principal intervention [36, 38, 40, 41, 43, 44, 47, 52, 53] (Table 2). None of these studies used a species-specific diagnostic test (S1 Table). One study of selective chemotherapy following ring-screening only measured the prevalence at follow-up in both the intervention and control groups [45] (Table 2). They calculated the prevalence ratio for the intervention vs control group, adjusted for age, sex, number of household residents and household clustering. This is the only study that used a species-specific diagnostic test but could not be included in the meta-analysis due to the different outcome measure (it did not measure cure rate or relative reduction in prevalence). The results of the 18 primary studies can be found in S2 Table.

To allow comparison between the different drugs and doses for preventive chemotherapy, a meta-analysis was conducted for all studies that measured cure rate and for which the effect could be attributed to one drug (Fig 3). This analysis showed a combined cure rate of 90.8% (95%CI 80.4–98.8%, 11 studies) but with substantial heterogeneity (Q = 129, P < 0.001, I$^2$ =

**Table 2. Characteristics of included studies that tested selective chemotherapy (n = 10).**

| Study ID and references | Country and year/s of study | Study type | Participants: number, age, specific characteristics | Intervention, drug and dose | Follow-up period | Outcomes |
|---|---|---|---|---|---|---|
| Bustos 2012 [36] | Peru 2004–2007 | BA | n = 69; all ages (mean 33, SD 15.7 years); diagnosed as *T. solium* taeniasis positive. | Selective chemotherapy: NICL 2 g (adults), 1.5 g (children > 35 kg), 1g (children 11 to 34 kg) | Immediately posttreatment and on days 1, 3, 7, 15, 30, and 90 post-treatment | • Infection rate with TS taeniasis—cure rate |
| de Kaminsky 1991 [38, 59] | Honduras NR | BA | n = 56; all ages (range 3–68 years); pregnant women excluded. | Selective chemotherapy: ALB 400 mg per day for 3 consecutive days | 24 h stools daily for 5 days after initiation of treatment, and then 60 and 90 days after treatment | • Infection rate with TS taeniasis—cure rate |
| Groll 1980 [40] | Various—majority of cases from Latin America NR | CBA | n = 42 for efficacy (IntA n = 33, IntB n = 9), n = 1046 for side-effects; all ages; persons with severe liver and intercurrent diseases, pregnant women and lactating women excluded. | Selective chemotherapy: IntA—PZQ 10mg/kg; IntB—PZQ 5mg/kg | 30, 60 and 90 days after treatment | • Infection rate with TS taeniasis—cure rate<br>• Risk of side-effects<br>• Observation time of side-effects |
| Jagota 1986 [41, 61–63] | India NR | CBA | n = 74 for efficacy (IntA n = 37, IntB n = 37), n = 480 for side-effects; all ages > 2 years (range 2–60 years); persons receiving or treated with an anthelmintic 7 days prior, with any acute illness, proteinuria or allergic disorders, and pregnant and lactating mothers excluded. | Selective chemotherapy: IntA—ALB 400 mg as a single dose; IntB—ALB 400 mg per day for 3 consecutive days as the only treatment or after failure with single dose ALB | 2 and 3 weeks after treatment, some at 3 months after treatment | • Infection rate with TS taeniasis—cure rate<br>• Risk of side-effects |
| Kumar 2014 [43] | India 2012–2013 | BA | n = 2732; all ages (mostly adult males, 76 females, including 6 children); persons suffering from diarrhea/dysentery excluded. | Selective chemotherapy plus education: PZQ 10 mg/kg[a] | 14–21 days post treatment | • Infection rate with TS taeniasis—cure rate<br>• Infection rate with STH—cure rate |
| Moreira 1983 [44] | Brazil NR | BA | n = 31; adults (17–60 years of age). | Selective chemotherapy: PZQ 10 mg/kg | 3 months after treatment | • Infection rate with TS taeniasis—cure rate<br>• Risk of side-effects |
| O'Neal 2014 [45] | Peru NR | CT | n = 1058 intervention, n = 753 control; all ages ≥2 years. | Int—Targeted (ring screening)[b] and then selective chemotherapy plus education: NICL 2g (>50kg), 1.5g (35–50 kg), 1g (11–34 kg)—if infection persisted after two weeks persons were re-treated with NICL and followed until the infection was cleared. Cont—education only | 16 months (after first treatment) 4 months (after final treatment) | • Infection rate with TS taeniasis–prevalence<br>• Porcine cysticercosis rate—seroincidence |
| Rim 1979 [47] | Korea NR | CBA | n = 53, ages 12–67 years; proven cases of *T. solium* infection. | Selective chemotherapy: IntA—PZQ 10 mg/kg; IntB—PZQ 5 mg/kg | 1–3, 30, 60 and 90 days after treatment | • Infection rate with TS taeniasis—cure rate<br>• Risk of side-effects |

(*Continued*)

**Table 2.** (Continued)

| Study ID and references | Country and year/s of study | Study type | Participants: number, age, specific characteristics | Intervention, drug and dose | Follow-up period | Outcomes |
|---|---|---|---|---|---|---|
| Taylor 1995 [52] | South Africa NR | BA | n = 200; children 4–6 years (mean 4.81, SD 0.34 years); 77% of study population had multiple parasite infestations. | Selective chemotherapy: PZQ 40 mg/kg[a] | 6 weeks (≈45 days) after treatment; and then another 6 weeks later checked for reinfestation, retreated and followed for another 3 weeks (≈21 days). | • Infection rate TS taeniasis—cure rate • Infection rate with STH—cure rate • Feasibility |
| Varma 1990 [53] | India NR | CBA | n = 74; NR | Selective chemotherapy: Int: NICL 2g; Cont: Mebendazole at 100mg, 200mg or 300mg per day for 3 days, Flubendazole at 200mg or 300mg per day for 3 days | 1–5, 30, 45, 60, 75 and 90 days after treatment | • Infection rate with TS taeniasis—cure rate • Risk of side-effects |

Abbreviations: ALB—albendazole; BA—before-after study; CBA—controlled before-after study; Cont—control; CT—controlled trial; Int—intervention; NICL—niclosamide; NR—not reported; PZQ—praziquantel; STH—soil transmitted helminths; TS—*Taenia solium*

[a] other drugs for different parasites, ALB 400mg for soil transmitted helminths

[b] pigs were screened for cysticercosis every 4 months for 12 months (4 occasions). Only residents living within a 100-meter ring surrounding the house where the heavily-infected pig was owned were screened and treated if positive for *Taenia solium*.

89%), suggesting that the pooled cure rate is probably not a robust estimate. The Doi plot (Panel A in S2 Fig) showed only minor asymmetry (LFK index -1.30), suggesting that publication bias is not an issue.

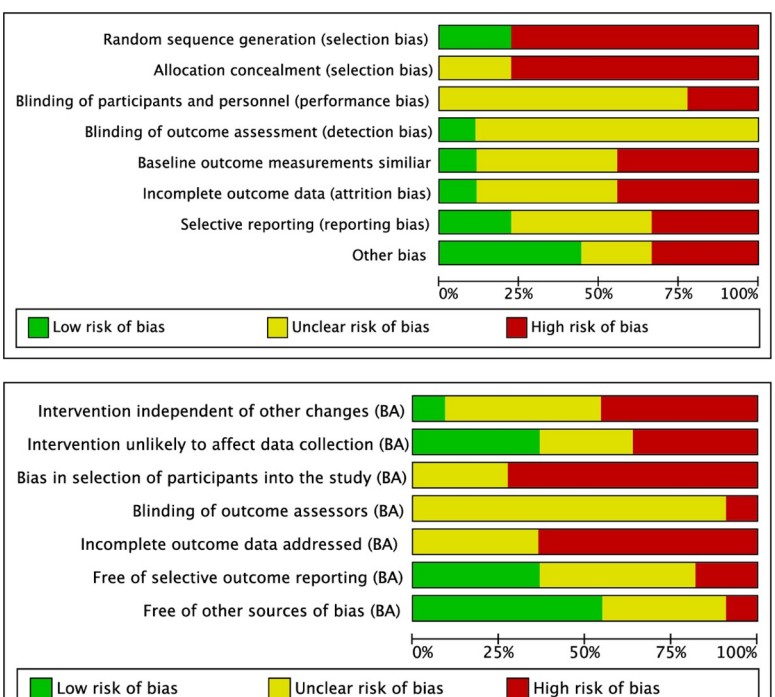

**Fig 2. Risk of bias graph: Review authors' judgements about each risk of bias item presented as percentages across all included studies.** Panel A, studies with a control group. Panel B, before-after studies.

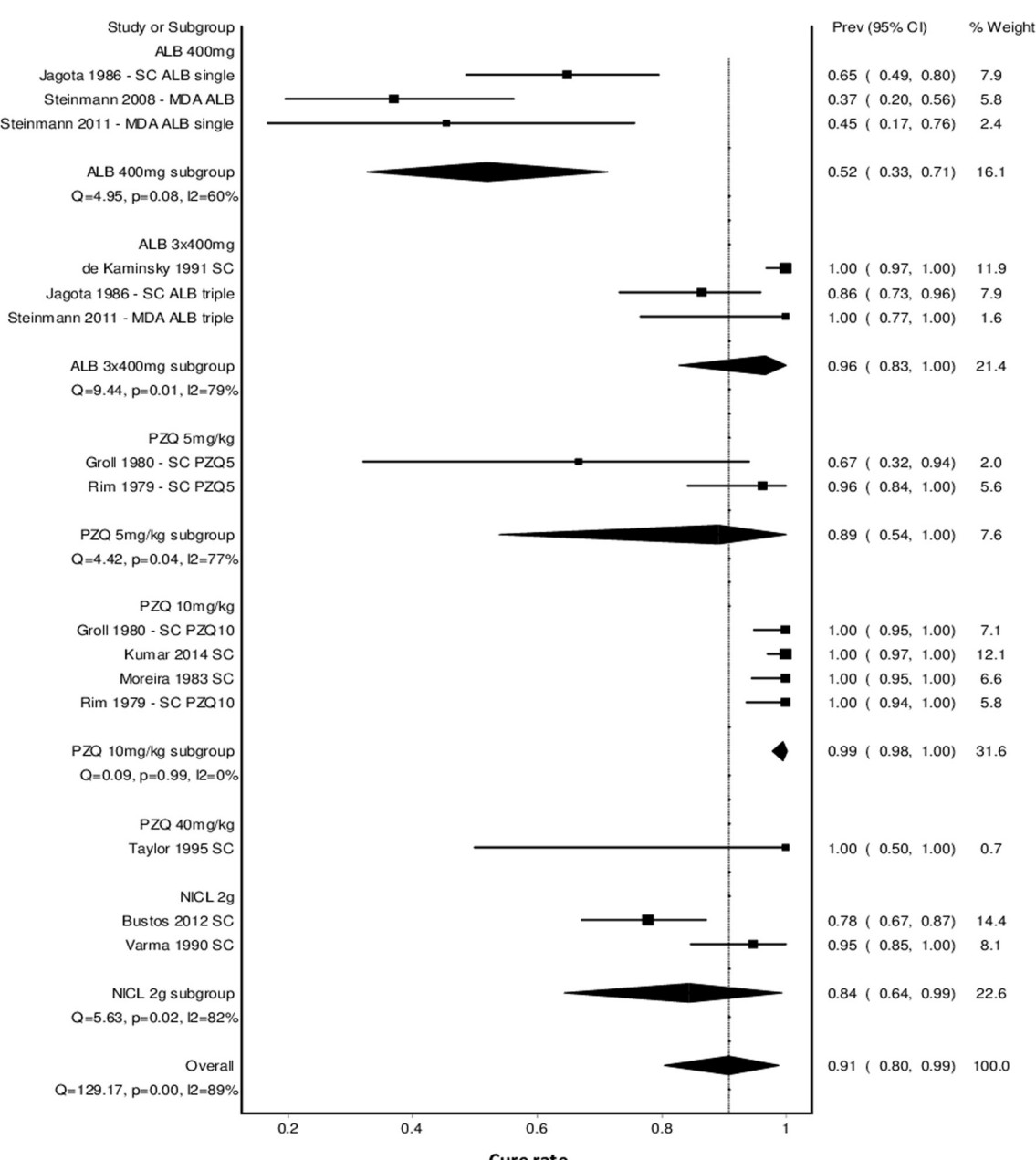

**Fig 3. Forest plot showing the effect of preventive chemotherapy with different drugs and doses on the cure rate for *T. solium* taeniasis.**

Subgroup analysis of drug and dose showed that a single dose of ALB 400mg had a significantly lower cure rate (52.0%, 95%CI 32.6–71.3%, 3 studies) than triple dose ALB 400mg given over 3 consecutive days (96.4%, 95%CI 82.8–100%, 3 studies) (Fig 3). There was no significant difference between PZQ at 5mg/kg body weight (89.0%, 95%CI 53.9–100%, 2 studies) and 10mg/kg (99.5%, 95%CI 97.7–100%, 4 studies), though PZQ at 10mg/kg tended to give better results (based on comparison of the pooled cure rates and 95%CI). PZQ at 40mg/kg was only tested in one study in preschool age children with only 3 children testing positive to *Taenia* spp., thus it is not possible to draw reliable conclusions at this dose. NICL was only tested as a

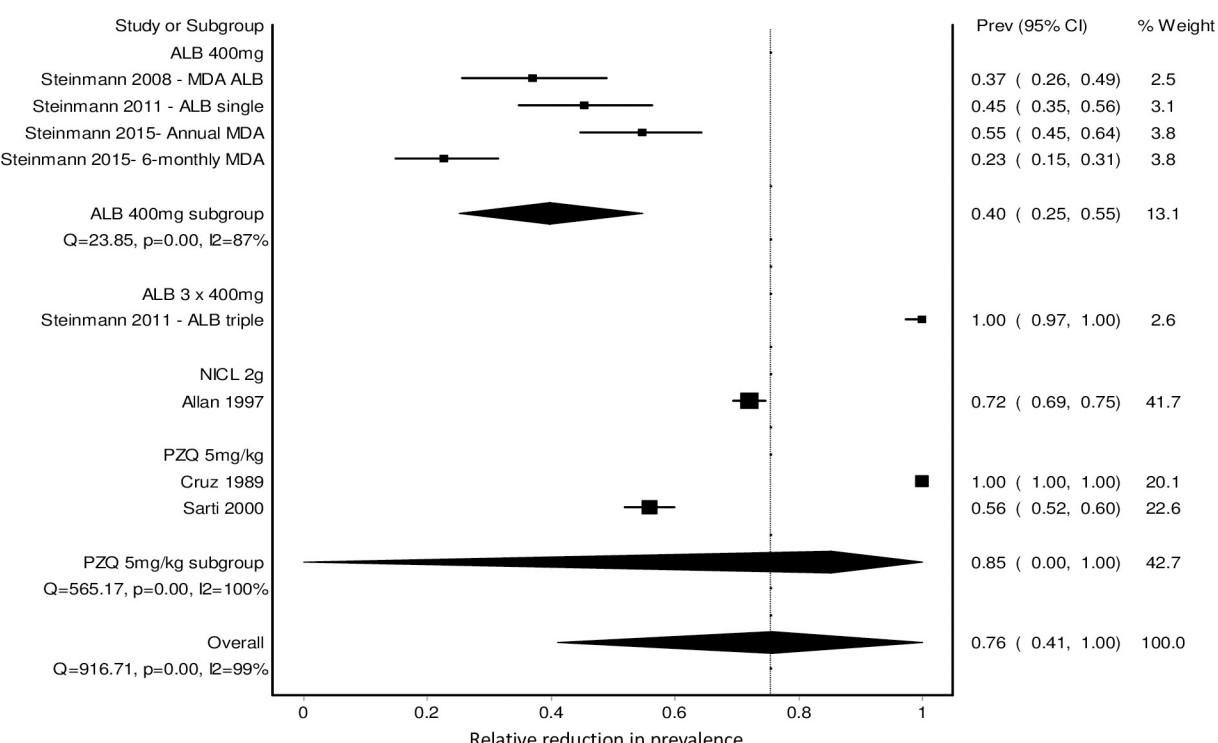

**Fig 4. Forest plot showing the effect of preventive chemotherapy with different drugs and doses on the relative reduction in prevalence for *T. solium* taeniasis.**

single dose of 2g (adjusted for children) with a combined cure rate of 84.3% (95%CI 64.4–99.3%, 2 studies). These combined cure rates (i.e. per subgroup) all had significant heterogeneity, except for PZQ 10mg/kg (Q = 0.09, p = 0.99, $I^2$ = 0%). There was no difference in cure rate between ALB 400mg given over 3 consecutive days, PZQ at 5 or 10mg/kg, and NICL at 2g, with significance judged by whether the confidence intervals overlapped or not (Fig 3).

The above analysis was complemented by considering the studies that used other measures, e.g. relative reduction in prevalence (Fig 4). For this analysis, we used the reduction in prevalence at the follow-up time listed in Table A in S2 Table. The follow-up time for these studies was most commonly 9–12 months after the baseline measure, though two studies tested at 1 month, one at 24 months and one at 6 and 42 months (with very similar results [48]) (Table A in S2 Table). The Doi plot (Panel B in S2 Fig) showed only minor asymmetry (LFK index -1.05), again suggesting that publication bias is not an issue. For ALB, one controlled before-after study of single dose ALB 400mg given as annual or 6-monthly MDA resulted in a relative reduction in prevalence of 55% (95%CI 45–64%) and 23% (95%CI 15–31%), respectively–measured one year after one annual MDA and six months after the second MDA (biannual MDA) [50]. In addition another two studies by Steinmann et al. published in 2008 [51] and 2011 [49] reported relative reduction in prevalence as well as cure rate so are also included in Fig 4. Overall, the relative reduction in prevalence for single dose ALB 400mg was 40% (95%CI 25–55%) (Fig 4). For PZQ, a dose of 5mg/kg was also tested in two before-after studies of MDA but showed very variable results as shown by the wide confidence interval around the combined result (relative reduction in prevalence = 85.3%, 95%CI 0–100%), with one study showing a relative reduction in prevalence of 100% [37] and the other 56% [48]. For NICL, an additional two studies tested a 2g dose. One of these was a before-after study of MDA with a

relative reduction in prevalence of 72% (95%CI 69–75%, [34] (shown in Fig 4)), and the other was a controlled trial of ring screening followed by selective chemotherapy, showing that the intervention resulted in a lower prevalence of *T. solium* at follow-up (adjusted prevalence ratio 0.28, 95%CI 0.08–0.91) compared to no intervention [45]. The follow-up time was 16 months after the first treatment, which was also four months after the final treatment. The baseline prevalence of taeniasis was not measured in this study so it was not possible to calculate the relative reduction in prevalence.

The certainty of the evidence for each of the above comparisons (triple dose ALB 400mg, single dose ALB 400mg, PZQ 10mg/kg, PZQ 5mg/kg, and NICL 2g each compared to no intervention) was very low, as judged using the GRADE approach. The certainty of the evidence was downgraded due to the use of study designs other than randomized control trials, for risk of bias, and for heterogeneity. There was only one study that did a direct comparison of cure rate for different drug doses using a randomized control trial design (triple dose ALB 400mg vs single dose ALB 400mg), resulting in a relative risk of 2.2 (95%CI 0.7–3.7) [49]. This finding was judged as having a low certainty of evidence, with the certainty of the evidence downgraded due to risk of bias and imprecision.

**Infection rate with soil-transmitted helminths.** Seven studies measured the infection rate with soil-transmitted helminths, of which three only measured the relative reduction in prevalence [39, 46, 50], two only measured the cure rate [43, 52] and two measured both [49, 51]. Results are presented in Table 3.

One study screened all inhabitants of a rural village in Mexico and treated individuals infected with soil-transmitted helminths with 400mg ALB and with taeniasis with 2g NICL [39]. Then all inhabitants were administered PZQ at 10mg/kg as part of MDA for taeniasis. Thus, it is not possible to attribute the reductions in prevalence of soil-transmitted helminths or taeniasis to a particular drug. Six of the studies tested the effect of ALB as a single dose of 400mg [43, 50–52], as a triple dose of 400mg (over 3 consecutive days) [46] or compared single vs triple dose ALB [49]. Three of these studies used ALB for MDA with the primary aim of controlling soil-transmitted helminths [49–51], with taeniasis being a secondary outcome, while one used ALB for MDA (along with other interventions in pigs) with the primary aim of controlling *T. solium* taeniasis [46]. The fifth study used ALB at 400mg for soil-transmitted helminths and PZQ at 10mg/kg body weight for taeniasis in a program of selective chemotherapy [43]. The sixth study used ALB at 400mg for children infected with soil-transmitted helminths and PZQ at 40mg/kg for taeniasis in a program of selective chemotherapy in preschool children [52]. While all studies achieved reductions in the prevalence of soil-transmitted helminths, along with reductions in *T. solium* taeniasis, none of the studies tested two drugs simultaneously. The randomized controlled trial comparing single vs triple dose (ALB) for MDA showed that triple dose is significantly more efficacious than single dose for both taeniasis and soil-transmitted helminths [49]. The certainty of evidence (GRADE) for this finding was judged as low.

**Risk of side-effects.** Risk of side-effects from NICL, PZQ or ALB, including seizures and severe headache, was measured in eleven studies– 6 in the context of MDA [35, 37, 46, 48, 49, 51] and 5 following selective chemotherapy [40, 41, 44, 47, 53] (Table 4). All three drugs and most doses (ALB single dose, ALB triple dose, PZQ 5mg/kg, PZQ 10mg/kg, NICL 2g) are represented by the studies. Most studies reported either no or only mild and transient side-effects within the first 3 days following drug administration. The only severe side-effects reported were one case of severe headache in a patient subsequently diagnosed as having neurocysticercosis [13, 48] and another case where a single patient had seizures following PZQ 5mg/kg that the authors suggest may not have been directly related to the treatment [37]. It is important to

**Table 3. Results for soil-transmitted helminths (n = 7).**

| Study ID, references & country | Study type | Participants: number, age, specific characteristics | Intervention, drug and dose | Results |
|---|---|---|---|---|
| Diaz Camacho 1991 [39] Mexico | BA | n = 339; all ages > 5 years; pregnant women and persons with liver cirrhosis or neurologic symptoms excluded. | Selective chemotherapy of cases: NICL 2g for taeniasis, ALB 400 mg per day for three consecutive days for *Hymenolepis nana*; ALB 400 mg for *ascaris*, *enterobius*, and *Trichuris* MDA—PZQ 10 mg/kg | Significant decreases in prevalences from baseline to follow-up were found for *T. trichiura* (prevalence decrease from 8.6% to 3.7%). Antihelminthic treatment decreased the overall prevalence of intestinal parasites from 69.2% at baseline to 37.5% at 1-year follow-up (n = 283 stool samples at follow-up). Relative reduction in prevalence = 45.8%. |
| Kumar 2014 [43] India | BA | n = 2732; all ages (mostly adult males, 76 females, including 6 children); persons suffering from diarrhea/dysentery excluded. | Selective chemotherapy plus education: PZQ 10 mg/kg for taeniasis, ALB (400mg) for *A. lumbricoides* and *T. trichiura*. *Hymenolepis nana*– 25 mg/kg body weight of PZQ, and the dose was repeated after one week. | The overall prevalence of intestinal parasitosis was found to be 49.38% (1349/2732). The prevalence of *Ascaris lumbricoides* was found to be the highest (46.88%), followed by *H. nana* (0.21%). The cure rate after one-time administration of recommended doses of anthelminthic drugs was found to be 66% for *Ascaris lumbricoides* and 100% for other parasites (n = 1349 positive at baseline). |
| Okello 2016 [14, 46, 54] Lao People's Democratic Republic | BA | n = 298; all ages > 6 years; persons with acute illness, pregnant and lactating women excluded. | MDA with ALB 400 mg per day for three consecutive days—twice 6 months apart (plus education and intervention in pigs—three times 6 months apart). | Relative reduction in prevalence of hookworm was 83.4% (after MDA1, n = 58 stool samples) and 84.5% (after MDA2, n = 48 stool samples), *A. lumbricoides* was 95.6% and 93.5% and *T. trichiura* was 69.2% and 61% after MDA1 and MDA2, respectively. The intensity of infection within the sampled population also decreased, with egg reduction rates of 94.4% (MDA1) and 97.8% (MDA2) for hookworm, 99.4% and 99.3% for *A. lumbricoides* and 77.2% and 88.5% for *T. trichiura*. During the 5-month inter-treatment interval between MDA1 and MDA2, an increase in STH prevalence was detected, with the overall prevalence reaching 62.1% of pre-MDA1 levels. Among the individual parasite species detected *A. lumbricoides* had the greatest increase in prevalence, reaching 74.8% of pre-MDA1 levels followed by 70.6% for *T. trichiura* and 48.4% for hookworm. |
| Steinmann 2008 [51] China | RCT | n = 66 ALB, n = 57 Tribendimidine; all ages (range 5–87 years); persons with acute illness, pregnant, or who had drunk alcohol on the day of treatment excluded. Area co-endemic for soil transmitted helminths. | Int—MDA with ALB 400 mg (≥ 15 years), 200 mg (5 to 14 years); Cont—MDA with Tribendimidine 400 mg (≥ 15 years), 200 mg (5 to 14 years) | ALB—Cure rate = 100% (50/50) for *A. lumbricoides*, 69.6% (32/46) for hookworm, 11.7% (7/60) for *T. trichiura*, and 30.8% (4/13) for *S. stercoralis*, respectively. TRB—Cure rate = 92.3% (36/39), 52.2% (24/46), 0% (0/48), and 46.2% (6/13), respectively. |

(*Continued*)

**Table 3.** (Continued)

| Study ID, references & country | Study type | Participants: number, age, specific characteristics | Intervention, drug and dose | Results |
|---|---|---|---|---|
| Steinmann 2011 [49, 64] China | RCT | n = 314; all ages ≥ 5 years; persons with chronic disease or other conditions likely to interfere with treatment, pregnant women, recent anthelminthic treatment excluded. Area co-endemic for soil transmitted helminths. | MDA IntA—single dose ALB 400 mg IntB—triple dose ALB 400 mg given over 3 consecutive days ContA—single dose mebendazole 500 mg ContB—triple dose mebendazole 500 mg given over 3 consecutive days | ALB—Single dose: Cure rate = 69.1% (95% CI: 55.2–80.9%), (17/55) for hookworm; 96.1% (95% CI: 89.1–99.2%), (75/78) for *A. lumbricoides*; 33.8% (95% CI: 22.6–46.6%), (22/65) for *T. trichiura*. ALB—Triple dose: cure rate = 92.0% (95% CI: 80.8–97.8%), (46/50) for hookworm; 96.8% (95% CI: 89.0–99.6%), (61/63) for *A. lumbricoides*; 56.2% (95% CI: 41.2–70.5%), (27/48) for *T. trichiura*. The efficacies of ALB and mebendazole were comparable. Triple dose treatment was significantly more efficacious than single dose treatment for hookworm and *T. trichiura* but not for *A. lumbricoides*: difference between cure rate for triple vs single dose for ALB was 22.9% for hookworm, p<0.01, 22.4% for *T. trichiura*, p<0.05, and 0.7% for *A. lumbricoides*, NS. |
| Steinmann 2015 [50] China | CBA | n = 760, 100 samples per village; all ages ≥ 2years; persons with acute or chronic illness, pregnant women excluded. Area co-endemic for soil transmitted helminths. | MDA—ALB 400mg IntA—Annual MDA; IntB—6-monthly MDA; IntC—6-monthly MDA + latrine construction + regular health education (all over a 3-year period). Then annual MDA by local village doctors for another 2-year period. | N≈100 in each group at each time point. Relative reduction in prevalence after 2 years: Hookworm—A) 93.3%, B) 84.3%, C) 72.7%; *A. lumbricoides*—A) 4.0%, B) 50.1%, C) 75.0%; *T. trichiura*—A) 27.7%, B) 19.7%, C) 41.5%; *S. stercoralis*—A) 33.1%, B) 59.0%, C) 5.4%. All three interventions significantly reduced the prevalence of *T. trichiura* and hookworm after 2 years and 5 years but not *S. stercoralis*. B and C significantly reduced the prevalence of *A. lumbricoides* after 2 years, while both A and C significantly reduced the prevalence of *A. lumbricoides* after 5 years. |
| Taylor 1995 [52] South Africa | BA | n = 200; children 4–6 years (mean 4.81, SD 0.34 years); 77% of study population had multiple parasite infestations. | Selective chemotherapy PZQ 40 mg/kg for taeniasis; ALB 400mg for soil transmitted helminths | Cure rate: *A. lumbricoides*—91.9% (113/123) after first treatment, 96.4% (27/28) after second treatment; *T. trichiura*—22.4% (28/125) after first treatment, 14.1% (12/85) after second treatment; *N. americanus*—89% (40/45) after first treatment, 100% (3/3) after second treatment. |

Abbreviations: ALB—albendazole; CI—confidence interval; MDA—mass drug administration; NICL—niclosamide; NS—not significant; PZQ—praziquantel; SD—standard deviation

note here that case reports were not included in the systematic review because they are a very low level of evidence of effect and chance could not be ruled out.

Generally the data on side effects was not recorded consistently (in the studies of MDA), except for the studies by Okello et al. [46] and Steinmann et al. [49]–both tested the effect of ALB triple dose (Table 4). While the study by Sarti et al. [48] report a house to house survey to measure side-effects in a supporting publication [13] it is not clear how long after the MDA this took place as it seems to have been prompted by the diagnosis of one case of neurocysticercosis identified following MDA with 5mg/kg PZQ.

**Risk of side-effects due to simultaneous medication.** No studies applied two different drugs simultaneously, thus there is no data available for this primary outcome. Where studies used more than one medication in the same individual, these were administered at distinct times.

**Table 4. Data extracted from included studies on the risk of side-effects (n = 11).**

| Study ID, references & country | Number of participants, Age group | Intervention | Risk of side-effects | Rate of neurological side-effects | Observation time of side-effects |
|---|---|---|---|---|---|
| Braae 2017 [35, 55–57, 60] Tanzania | n>3000 School aged children (4–15 years) | Children—4–15 years MDA: PZQ—40 mg/kg; Selective chemotherapy: NICL - 50mg/kg or PZQ - 10mg/kg for Adults—> 15 years Selective chemotherapy: NICL - 2g or PZQ - 10mg/kg | Participants were asked to report side-effects and a form provided for this. No side-effects were reported during the study period from any of the schools or individuals treated during the study. | | |
| Cruz 1989 [37, 58] Ecuador | n = 10,173 All ages | MDA PZQ—5 mg/kg body weight | "Side-effects relating to the treatment were not recorded consistently and could therefore not be analyzed statistically. However, to the best of our knowledge, all side-effects were transient and mild; more serious was a case of seizures and also a case of dysentery that may not have been related directly to the treatment." | One case of seizures. | |
| Groll 1980 [40] Various countries | n = 1046[a] All ages | Selective chemotherapy A. PZQ—5 mg/kg B. PZQ—10 mg/kg | For cestode infections in general: "The tolerance of the drug was good. Of totally 1046 patients treated with Praziquantel, 47 showed subjective symptoms. Headache, dizziness, abdominal discomfort and nausea were the main complaints, some of them by the same patients. Two skin rashes were observed within 24 h after treatment. Special medical care was not necessary for any of the reported adverse reactions." Laboratory examinations did not show any clinically relevant alterations from normal values. | | "These symptoms persisted some minutes up to 3–4 h." |
| Jagota 1986 [41, 61–63] India | n = 480[b] All ages (> 2 years) | Selective chemotherapy A. ALB 400 mg as a single dose B. ALB 400 mg per day for 3 consecutive days | ALB was well tolerated, and no changes were observed in laboratory test results. 5.8% (28/480) of patients complained of various adverse reactions, mostly gastrointestinal disorders. 1 had headache. All adverse reactions were mild. The drug was well tolerated. Note: all patients (not just *Taenia* spp.) were included in the assessment of side-effects. | | |

(*Continued*)

**Table 4.** (Continued)

| Study ID, references & country | Number of participants, Age group | Intervention | Risk of side-effects | Rate of neurological side-effects | Observation time of side-effects |
|---|---|---|---|---|---|
| Moreira 1983 [44] Brazil | n = 31 Adults (17–60 years) | Selective chemotherapy PZQ—10 mg/kg tablets | Eight potential side effects events attributable to the anthelmintic were recorded (assumed by the systematic reviewers to be within two hours of treatment, which was the observation time): headache, intestinal colic, diarrhea, nausea, feeling of fatigue, drowsiness and dizziness. These had short duration and regressed spontaneously. | | |
| Okello 2016 [14, 46, 54] Lao People's Democratic Republic | n = 298 All ages (> 6 years) | MDA ALB—400 mg per day for three consecutive days | During each MDA round, local government medical personnel visited all households for three consecutive days in order to administer anthelmintic tablets. This protocol enabled medical staff to assess patient health after the previous day's medication in order to effectively monitor and address any adverse reactions. "No extreme adverse reactions occurred. Mild reactions including headaches, vertigo, breathing difficulties and gastrointestinal discomfort were efficiently addressed by the medical teams." Bardosh et al. 2014 "Although no Extreme Adverse Reactions (EARs) were reported, likely due to the lower drug dosages given over three consecutive days, abdominal pain, headache, constipation, tiredness, coughing, difficulty breathing, bloating, high heart rate and vertigo were all reported." | | |
| Rim 1979 [47] Korea | n = 53 Age 12–67 years | Selective chemotherapy A. PZQ—5 mg/kg B. PZQ—10 mg/kg | A. 48.2% (13/27) had side-effects—Abdominal pain (11), soft stool or diarrhea (4), dizziness (2), nausea (1), headache (1), urticaria (1); B. 53.9% (14/26) had side-effects—Abdominal pain (13), soft stool or diarrhea (6), dizziness (3), nausea (0), headache (0), urticaria (1); | | |
| Sarti 2000 [13, 48] Mexico | n = 2452 All ages (> 4 years) | MDA PZQ—5 mg/kg body weight | "in general, side-effects were low (1.9% developed headache, nausea, vomiting or abdominal pain)" [48]. Following a house to house survey: "Another 31 others were found, among the 2452 who took the taeniacidal dose of praziquantel (1.3%), who developed headaches (for 1–10 days) within 3 days of taking the drug" [13]. | 1 case of neurocysticercosis diagnosed with MRI following severe headache for about 10 days within 24 hours of taking the PZQ [13] | |

(*Continued*)

**Table 4.** (*Continued*)

| Study ID, references & country | Number of participants, Age group | Intervention | Risk of side-effects | Rate of neurological side-effects | Observation time of side-effects |
|---|---|---|---|---|---|
| Steinmann 2008 [51] China | n = 66 All ages (5–87 years) | MDA ALB—200mg (5 to 14 years), 400mg (≥ 15 years) | "No adverse events were mentioned by participants treated with single dose oral ALB." | | |
| Steinmann 2011 [49, 64] China | n = 314 All ages ≥ 5 years | MDA A. single dose ALB (400 mg), n = 82 B. triple dose ALB (3x400 mg, given over 3 consecutive days), n = 68 C. single dose mebendazole (400 mg) D. triple dose mebendazole (3x400 mg, given over 3 consecutive days). | 36 hours after the first dosing participants were actively solicited to report any potential adverse events. Reported health problems were classified by the study physician and graded by severity according to a pre-defined scale. Thirteen study participants (4.1%) reported between one and five adverse events following drug administration . . . Four of these individuals were treated with a single dose (3 with mebendazole, 1 with ALB) while the remaining nine were treated with triple mebendazole (n = 5) or triple ALB (n = 4). Adverse events included headache (n = 3; all mebendazole), abdominal cramps (n = 3; 2 mebendazole, 1 ALB) and the closely related "full stomach" (n = 2; mebendazole), and waist pain (n = 1; ALB). Two individuals each reported vomiting, including production of A. lumbricoides worms (1 ALB, 1 mebendazole), diarrhea (2 mebendazole), fatigue (1 ALB, 1 mebendazole), and chills (2 mebendazole). Vertigo (ALB), throat pain (ALB), fever (mebendazole), and a swollen face (mebendazole) were each reported once. None of the study participants requested medical interventions as adverse events were mild and self-limiting. More women than men reported adverse events (10 vs 3, p = 0.046)). There was no significant association between the report of adverse events and age, drug, or number of treatments according to the Fisher's exact test. | | Of the adverse events reported following drug administration, most occurred in the morning of the third drug distribution day (about 12 hours after the administration of the second dose, if given) and upon active questioning. |
| Varma 1990 [53] India | n = 38 NR | Selective chemotherapy NICL—2g in one day—2 x 500mg tablets, then repeated after one hour | No side effects were observed in any of the patients treated. | | |

Abbreviations: ALB—albendazole; MDA—mass drug administration; NICL—niclosamide; PZQ—praziquantel

[a] side-effects reported for all participants in the trials—cestode infections in general, all treated with PZQ

[b] side-effects reported for all participants in the trials—nematode and cestode infections in general, all treated with ALB

**Observation time of side effects due to NICL, ALB or PZQ.** No studies reported this outcome and those that did measure side-effects generally reported side-effects occurring within the first 3 days following drug administration (Table 4). However, authors of two studies [40, 49] did make specific comments on the timing of the side effects [40, 49] (Table 4).

**Cost, cost-effectiveness, feasibility, values and preferences of participants, and impact on equity.** No studies measure the cost-effectiveness of the different drugs but one of the included studies did report the cost of treatment [37] (S3 Table). None of the studies specifically aimed to measure the feasibility of MDA but four made comments (generally in the discussion section of the paper) regarding aspects of feasibility [37, 42, 46, 52] (S3 Table). Four studies commented on aspects related to values and preferences of participants [37, 42, 46, 49, 54]. No studies measured the impact on equity of MDA but one study did comment on the distribution of taeniasis (at baseline) according to economic conditions [37] (S3 Table).

**Porcine cysticercosis.** Six studies of MDA and one study of ring-screening and selective chemotherapy measured the secondary outcome of porcine cysticercosis prevalence or seroprevalence [34, 35, 37, 39, 42, 45, 48].

## Discussion

Due to the potential for *T. solium* to cause neurocysticercosis [3], it is important to implement strategies for the control of this parasite. One such strategy is preventive chemotherapy for *T. solium* taeniasis using MDA in endemic populations. However, such programs need to be informed by evidence of the best drug and dose in terms of efficacy and side-effects. The present systematic review was conducted to provide such evidence and inform the development of guidelines for preventive chemotherapy. It is the first systematic review conducted on this topic.

In relation to efficacy, analyses of drug and dose showed that PZQ 10mg/kg, triple dose ALB 400mg (400mg per day for three consecutive days) and NICL 2g resulted in better cure rates for *T. solium* taeniasis (99.5%, 96.4% and 84.3%, respectively) than PZQ 5mg/kg or single dose ALB 400mg (89.0% and 52.0%, respectively). Further, the cure rate between PZQ 10mg/kg, triple dose ALB 400mg and NICL 2g showed no statistically significant difference. However, it should be noted that the lack of statistical significance does not mean that there is no difference but could be due to the small number of studies. There was a tendency, however, for PZQ 10mg/kg and triple dose ALB 400mg to report higher cure rates than NICL 2g. It is important to note that most of these findings have a low certainty of evidence due to high risk of bias in individual studies and heterogeneity in combined estimates.

Those studies included in the systematic review that used relative reduction in prevalence of *T. solium* taeniasis (mostly studies of MDA) generally support the above findings but show lower efficacy due to the influence of other factors such as population coverage, timing of follow-up and sampling. Two studies of MDA using a single dose of NICL 2g were conducted as part of a series of studies in Tumbes, Peru, over 3 phases–both reported cure rates [67, 74]. They did not meet the inclusion criteria for this systematic review due to insufficient information on methods and results. Nonetheless, the results are pertinent here. In phase 1 of these studies, Garcia et al. found a cure rate of 63.2% (24/38) at 2 weeks after mass treatment with NICL 2g, as assessed by coproantigen detection plus stool microscopy [74]. In phase 3 of these studies, Gamboa et al. found a cure rate of 71.9% (151/210) at 1 month following mass treatment with NICL 2g [67]. These cure rates are lower than those found in studies included in the systematic review (84.3%) and support the tendency for lower efficacy of NICL 2g in comparison to triple dose ALB 400mg and PZQ 10mg/kg.

In relation to adverse effects, most studies reported either no or only mild and transient side-effects within the first 3 days following drug administration for all drugs and doses. However, side-effects due to the different drugs and doses were generally not reported consistently, with the exception of two studies that tested the effect of triple dose ALB 400mg [46, 49] that showed only mild, transient side-effects. It is important to note that there were two reports of neurological side-effects following PZQ 5mg/kg, including severe headaches in one case of undiagnosed neurocysticercosis [13] and another case of seizures that the authors suggest may not have been directly related to the treatment [37]. These reports, along with 2–3 other case reports of neurological side-effects following administration of PZQ for *Taenia saginata* and/ or *Hymenolepis nana* [75–77] have led to some concern about the safety of PZQ in areas endemic for cysticercosis. However, perspective is warranted here as case reports are a very low level of evidence of effect and chance cannot be ruled out. In fact, in the WHO/FAO/OIE 2005 Taeniasis/Cysticercosis Guidelines [12] the authors note: *"this adverse effect is not a reason to cease using praziquantel in mass treatment interventions because sooner or later some asymptomatic cases of neurocysticercosis may become symptomatic due to the natural course of infection; treatment with praziquantel may accelerate this process on sporadic occasions. The use of low but still effective doses of praziquantel decreases the risk of such complications."*

Given the limited evidence on adverse effects found in the systematic review, a supplementary (though not systematic) search was conducted for studies reporting side-effects from treatment with NICL, PZQ or ALB for taeniasis (but not in sufficient detail to be included in the systematic review) or for other parasites. A summary of the 12 included studies is shown in S4 File. In relation to PZQ, this drug is given in doses of 40mg/kg for MDA against schistosomiasis, which is a dose 4–8 times greater than that given for MDA for *T. solium* taeniasis. Two studies found in the supplementary search monitored side-effects following MDA with PZQ 40mg/kg in *T. solium* endemic countries and found only mild and transient side-effects, which gradually resolved within 24 hours [78, 79]. No cases of neurological symptoms being triggered suggestive of neurocysticercosis were reported in these studies. Further, unpublished data from Madagascar, where PZQ 10mg/kg was used in around 70,000 people per year for 3 years for MDA for *T. solium* taeniasis, found no major side-effects. Follow-up of side-effects was for 5–7 days, both active and passive, and was done by Community Health Agents who live in the villages where they distributed the medicines (Sylvia Ramiandrasoa personal communication).

In relation to ALB, this is a drug that is used extensively at a single dose of 400mg for MDA against soil-transmitted helminths, with no serious adverse events reported [15]. A comprehensive, multi-country randomized placebo-controlled trial (n = 870, ages 3–79 years) found no significant difference in the number of side-effects between ALB 400mg (children <12 years were given 200mg) and placebo [80]. Subjects were recruited from France, Morocco, Mali, Senegal, Nigeria, Central African Republic, Kenya, Brazil, Peru, Mexico and the Philippines, which include countries endemic for *T. solium*.

A key strength of this review was the use of high-quality systematic review methods [16] that included the use of two reviewers for all stages of the review, i.e. screening of title/abstracts, study selection, data extraction and quality assessment. In addition, the search strategy was very comprehensive, including eleven electronic databases as well as contact with experts and specific searches for grey literature.

This review is limited by the study design and high risk of bias of all included studies. Also, the diagnostic tests used for measurement of *T. solium* taeniasis were not species specific, except in one included study [45]. This is understandable given the challenges that researchers face when carrying out studies on *T. solium*, often in rural, remote areas; the relatively recent development of species-specific tests; and the technical complexity and cost of implementing highly specific diagnostic tools in field conditions. Further, all but one of the studies that

measured cure rate relied on microscopy for diagnosis, which is known to lack sensitivity, thus likely to lead to an overestimation of cure rates [36]. Any differences that may exist in the sensitivity of the different species causing taeniasis in humans to PZQ, NICL or ALB could potentially contribute to inaccuracy in the cure rates interpreted for *T. solium*, specifically in those studies where a species-specific diagnostic test was not used. A further limitation, especially in the studies of MDA, was variation in the time to follow-up after treatment; longer periods risking the inclusion of new cases of infection might have been inferred as treatment failures–resulting in lower measured effectiveness. A limitation to the evaluation of side effects may be the duration patients were followed after treatment. At this time there is no consensus about the most appropriate time that would be required.

In conclusion, PZQ 10mg/kg, NICL 2g, and triple dose ALB 400mg can be considered for use in mass drug administration programs for the control of *T. solium* taeniasis. Given that there are some general concerns about the safety of PZQ and ALB in *T. solium* endemic areas, those drugs should not be administered to people with clinical signs compatible with NCC. Active monitoring of side-effects following MDA with PZQ 10mg/kg and triple dose ALB 400mg may be warranted to allay concerns and to add to the existing evidence regarding safety of these two drugs.

Future efficacy research should focus on the conduct of high quality randomized controlled trials of the three different drugs and varying doses (triple dose ALB 400mg, PZQ 5 vs 10mg/kg, NICL 2g)–both in comparison to placebo and in head-to-head trials. Care should be taken to ensure that the trials include (and report) random sequence generation, allocation concealment, blinding of participants, personnel and outcome assessors, adequately address incomplete outcome data, have a pre-registered/published protocol and avoid conflicts of interest [81]. Further, a higher dose or repeated dose of NICL should be tested to determine if it leads to improved efficacy. Where available, a *T. solium* species-specific diagnostic test should be used that also has high sensitivity.

## Supporting information

**S1 Checklist. PRISMA checklist.**
(DOC)

**S1 File. Search strategies and results.**
(DOCX)

**S2 File. Risk of bias assessment.**
(DOCX)

**S3 File. List of excluded studies with reason for their exclusion.**
(DOCX)

**S4 File. Side-effects reported in studies found through supplementary searching of treatment with albendazole, niclosamide or praziquantel of other parasites.**
(DOCX)

**S1 Fig. Risk of bias graphs: Review authors' judgements about each risk of bias item for each included study.**
(DOCX)

**S2 Fig. Doi plots of publication bias for the effect of preventive chemotherapy with different drugs and doses for *Taenia solium* taeniasis.** Panel A, meta-analysis of cure rate. Panel B,

meta-analysis of relative reduction in prevalence.
(DOCX)

**S1 Table. Strategies used to diagnose *Taenia solium* taeniasis.**
(DOCX)

**S2 Table. Infection rate with *Taenia solium* taeniasis.** A. Results of the studies that tested mass drug administration (with or without selective chemotherapy). B. Results of the studies that tested selective chemotherapy .
(DOCX)

**S3 Table. Data extracted from included studies on the outcomes: costs, cost-effectiveness, feasibility, values and preferences of participants, and impact on equity.**
(DOCX)

## Acknowledgments

We thank the members of the PAHO/WHO Guideline Development Group for their feedback on the systematic review protocol and an earlier version of this systematic review.

## Disclaimer

Some authors are staff members of the Pan American Health Organization. The authors alone are responsible for the views expressed in this publication, and they do not necessarily represent the decisions or policies of the Pan American Health Organization.

## Author Contributions

**Conceptualization:** Michelle M. Haby, Leopoldo A. Sosa Leon, Ana Luciañez, Ruben Santiago Nicholls, Ludovic Reveiz, Meritxell Donadeu.

**Data curation:** Michelle M. Haby, Leopoldo A. Sosa Leon.

**Formal analysis:** Michelle M. Haby, Leopoldo A. Sosa Leon.

**Funding acquisition:** Ana Luciañez, Ruben Santiago Nicholls.

**Methodology:** Michelle M. Haby, Leopoldo A. Sosa Leon, Ana Luciañez, Ruben Santiago Nicholls, Ludovic Reveiz, Meritxell Donadeu.

**Project administration:** Ana Luciañez, Ruben Santiago Nicholls.

**Supervision:** Ana Luciañez, Meritxell Donadeu.

**Validation:** Meritxell Donadeu.

**Visualization:** Michelle M. Haby, Leopoldo A. Sosa Leon.

**Writing – original draft:** Michelle M. Haby, Leopoldo A. Sosa Leon, Meritxell Donadeu.

**Writing – review & editing:** Michelle M. Haby, Leopoldo A. Sosa Leon, Ana Luciañez, Ruben Santiago Nicholls, Ludovic Reveiz, Meritxell Donadeu.

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
