## [Decision Letter · Decision Letter 0]

11 Sep 2019

Dear Dr. Haby:

Thank you very much for submitting your manuscript "Systematic review of the effectiveness of selected drugs for preventive chemotherapy for Taenia solium taeniasis" (PNTD-D-19-01307) for review by PLOS Neglected Tropical Diseases. Your manuscript was fully evaluated at the editorial level and by independent peer reviewers. The reviewers appreciated the attention to an important topic but identified some aspects of the manuscript that should be improved.

We therefore ask you to modify the manuscript according to the review recommendations before we can consider your manuscript for acceptance. Your revisions should address the specific points made by each reviewer.

(1) A letter containing a detailed list of your responses to the review comments and a description of the changes you have made in the manuscript.

(2) Two versions of the manuscript: one with either highlights or tracked changes denoting where the text has been changed (uploaded as a "Revised Article with Changes Highlighted" file ); the other a clean version (uploaded as the article file).

(3) If available, a striking still image (a new image if one is available or an existing one from within your manuscript). If your manuscript is accepted for publication, this image may be featured on our website. Images should ideally be high resolution, eye-catching, single panel images; where one is available, please use 'add file' at the time of resubmission and select 'striking image' as the file type. 

Please provide a short caption, including credits, uploaded as a separate "Other" file. If your image is from someone other than yourself, please ensure that the artist has read and agreed to the terms and conditions of the Creative Commons Attribution License at http://journals.plos.org/plosntds/s/content-license (NOTE: we cannot publish copyrighted images). 

(4) Appropriate Figure Files 

Please remove all name and figure # text from your figure files upon submitting your revision. Please also take this time to check that your figures are of high resolution, which will improve both the editorial review process and help expedite your manuscript's publication should it be accepted. Please note that figures must have been originally created at 300dpi or higher. Do not manually increase the resolution of your files. For instructions on how to properly obtain high quality images, please review our Figure Guidelines, with examples at: http://journals.plos.org/plosntds/s/figures

While revising your submission, please upload your figure files to the Preflight Analysis and Conversion Engine (PACE) digital diagnostic tool, https://pacev2.apexcovantage.com/ PACE helps ensure that figures meet PLOS requirements. To use PACE, you must first register as a user. Then, login and navigate to the UPLOAD tab, where you will find detailed instructions on how to use the tool. If you encounter any issues or have any questions when using PACE, please email us at figures@plos.org.

We hope to receive your revised manuscript by Nov 10 2019 11:59PM. If you anticipate any delay in its return, we ask that you let us know the expected resubmission date by replying to this email.

To submit your revised files, please log in to https://www.editorialmanager.com/pntd/

Sincerely,

Aysegul Taylan Ozkan, Ph.D., M.D.

Associate Editor

Jennifer Keiser

Deputy Editor

Reviewer's Responses to Questions

Key Review Criteria Required for Acceptance?

Methods

-Are the objectives of the study clearly articulated with a clear testable hypothesis stated?

-Is the study design appropriate to address the stated objectives?

-Is the population clearly described and appropriate for the hypothesis being tested?

-Is the sample size sufficient to ensure adequate power to address the hypothesis being tested?

-Were correct statistical analysis used to support conclusions?

-Are there concerns about ethical or regulatory requirements being met?

Reviewer #1: see below

Reviewer #2: The study, considering its type, follow clear and concrete objectives and the methods are in good articulation with these objectives . The only thing that I would add to the explanation of the methods used is the qualifications, competences and language skills for the two independent reviewers

Reviewer #3: The methodology that was used for this systematic review is of good quality; among other things, PRISMA checklist and flow diagram are included; 2 reviewers performed the searches and screening and several electronic databases were used.

Results

-Does the analysis presented match the analysis plan?

-Are the results clearly and completely presented?

-Are the figures (Tables, Images) of sufficient quality for clarity?

Reviewer #1: see below

Reviewer #2: Results are presented in a clear and complete form, and the tables show data in a suitable way

Reviewer #3: The data are presented well and the authors are open about the papers/data that were used.

Conclusions

-Are the conclusions supported by the data presented?

-Are the limitations of analysis clearly described?

-Do the authors discuss how these data can be helpful to advance our understanding of the topic under study?

-Is public health relevance addressed?

Reviewer #1: see below

Reviewer #2: Conclusions are straight and supported by the evidence shown. Limitations or potential bias are clearly stated. The public health relevance of the study and its results are addressed correctly

Reviewer #3: The authors are generally clear about the limitations and uncertainties that remain. The general picture and public health relevance is clearly addressed.

Editorial and Data Presentation Modifications?

Reviewer #1: see below

Reviewer #2: Use always italics when writing scientific names, even in the abstract, at the beginning of the manuscript

In line 36: It would be recommendable to give more details on PROSPERO

In line 55: Consider the inclusion of a comma: ....are considered, by the WHO, to be neglected....

In line 59: I do not consider that it is totally true that after years of treating people it is not known the best drug and the most appropriate dose are completely, I think that the statement could be changed to reflect in a better way what is the status on this area and what it still to be explore and assessed 

In lines 86-87: I think that the redundant reference to seizures could be modified to avoid repetitions of the same idea

In line 103: I do not find useful the reference to "traditional medicine" as the goal of the work has nothing to do with this topic

In line 107: Correct the term "inn", it should read "in"

In line 115: Avoid using italics in writing "spp"

In line 138: Include the term PROSPERO to be correspondent to the abstract and summary 

In line 170: I am not fully convinced of the use for the term "copro-DNA", perhaps it could be changed to "molecular techniques" or "use of genomic markers" or something similar

Reviewer #3: (No Response)

Summary and General Comments

Reviewer #1: I enjoyed very much reading this manuscript. I have several comments and recommendations that I hope the authors will follow. 

In lines 39 - 41 praziquantel should go before albendazole, so that efficacies can be listed from higher to lower, the same is true for lines 41-42, for figure 3 and for lines 493-494

Line 62 where should be when

Line 184 are should be changed for have

Line 266 I think that in sufficient should be insufficient

Line 272 I think the search performed by the authors deserves an explanation of the reasons why 3299 records were removed

Line 357 the authors should give a valid explanation to the fact that the only study that used a species-specific diagnostic test was not included because of "different outcome measure"

Line 373 could the authors please specify why PZQ at 10 mg/ml "tended to give better results"

Lines 415 and 416 have duplicated information, I suggest eliminating that of line 416.

The first paragraph of the discussion (lines 486-491) does not mention the word taeniasis and I think this is an important absence.

Line 537 Schistosomiasis should be without a capital letter

Line 563 "one included study (45), but in line 357 it states that this study was not included

I don't agree with the authors that conclude with a recommendation (lines 585-593) that more studies should be done to evaluate triple dose ALB 400mg, PZQ 5 vs 10mg/kg, NICL 2g; they convinced me that all, except PZQ at 5mg are good choices and, in endemic areas, niclosamide would be selected.

I think that references 37 and 58 are the same study; the first one is described in English, the second one in Spanish.

Table 3 lacks countries; it would be useful to include this information

Table 4 last line of the first column, the word India should be added

Figure 1 first square of records excluded, should have a brief explanation

Figure 1 second square, the explanation of full text articles "with reasons", is absurd, or you add an adequate explanation or you delete the one used

Figure 2 panel B, third line says slection, should be selection

Congratulations

Reviewer #2: In general this manuscript constitutes a very interesting and well written work, with clear objectives, which are scientifically justified, and clear results. Only some minor modifications suggested could be incorporated to further improve the quality of the document

Reviewer #3: L250: ‘infection rate’ doesn’t seem to be the correct term here since you mention that cure rate/reduction is usually used.

L457: so what are the conclusions of these studies [46 and 49]

L493-500: not finding a difference doesn’t mean there is no difference (L496-497). The number of studies was extremely low to do a proper evaluation. This should be clearly mentioned. 

L550-552: this is not very clear. There was no difference for children < 12="" _years2c_="" but="" in="" l552="" it="" is="" mentioned="" ages="" 3-79="" years.="" figure="" _13a_="" the="" additional="" records="" are="" 23="" _figure2c_="" seems="" to="" be="" different="" from="" number="" s1="" file.="" _22_full-text="" articles="" _excluded2c_="" with="" _reasons22_3a_="" which="" _reasons3f_="" include="" reasons.="" diagnosis="" of="" taeniosis="" very="" difficult.="" this="" should="" emphasized="" more="" manuscript="" as="" a="" major="" limitation="" for="" proper="" evaluation="" drugs="" effectiveness.="" methodology="" that="" was="" used="" _figures2f_tables="" where="" possible.="" _l592-5933a_="" test="" would="" you="" _suggest3f_="" --------------------="" plos="" authors="" have="" option="" publish="" peer="" review="" history="" their="" article="" />what does this mean?). If published, this will include your full peer review and any attached files.

Do you want your identity to be public for this peer review? For information about this choice, including consent withdrawal, please see our Privacy Policy.

Reviewer #1: Yes: Ana Flisser

Reviewer #2: Yes: Daniel A. Zarate-Rendon

Reviewer #3: No

---

## [Decision Letter · Decision Letter 1]

14 Oct 2019

Dear Dr. Haby:

Thank you very much for submitting your manuscript "Systematic review of the effectiveness of selected drugs for preventive chemotherapy for Taenia solium taeniasis" (PNTD-D-19-01307R1) for review by PLOS Neglected Tropical Diseases. Your manuscript was fully evaluated at the editorial level and by independent peer reviewers. The reviewers appreciated the attention to an important topic but identified some aspects of the manuscript that should be improved.

We therefore ask you to modify the manuscript according to the review recommendations before we can consider your manuscript for acceptance. Your revisions should address the specific points made by each reviewer.

(1) A letter containing a detailed list of your responses to the review comments and a description of the changes you have made in the manuscript.

(2) Two versions of the manuscript: one with either highlights or tracked changes denoting where the text has been changed (uploaded as a "Revised Article with Changes Highlighted" file ); the other a clean version (uploaded as the article file).

(3) If available, a striking still image (a new image if one is available or an existing one from within your manuscript). If your manuscript is accepted for publication, this image may be featured on our website. Images should ideally be high resolution, eye-catching, single panel images; where one is available, please use 'add file' at the time of resubmission and select 'striking image' as the file type. 

Please provide a short caption, including credits, uploaded as a separate "Other" file. If your image is from someone other than yourself, please ensure that the artist has read and agreed to the terms and conditions of the Creative Commons Attribution License at http://journals.plos.org/plosntds/s/content-license (NOTE: we cannot publish copyrighted images). 

(4) Appropriate Figure Files 

Please remove all name and figure # text from your figure files upon submitting your revision. Please also take this time to check that your figures are of high resolution, which will improve both the editorial review process and help expedite your manuscript's publication should it be accepted. Please note that figures must have been originally created at 300dpi or higher. Do not manually increase the resolution of your files. For instructions on how to properly obtain high quality images, please review our Figure Guidelines, with examples at: http://journals.plos.org/plosntds/s/figures

While revising your submission, please upload your figure files to the Preflight Analysis and Conversion Engine (PACE) digital diagnostic tool, https://pacev2.apexcovantage.com/ PACE helps ensure that figures meet PLOS requirements. To use PACE, you must first register as a user. Then, login and navigate to the UPLOAD tab, where you will find detailed instructions on how to use the tool. If you encounter any issues or have any questions when using PACE, please email us at figures@plos.org.

We hope to receive your revised manuscript by Dec 13 2019 11:59PM. If you anticipate any delay in its return, we ask that you let us know the expected resubmission date by replying to this email.

To submit your revised files, please log in to https://www.editorialmanager.com/pntd/

Sincerely,

Aysegul Taylan Ozkan, Ph.D., M.D.

Associate Editor

Jennifer Keiser

Deputy Editor

Reviewer's Responses to Questions

**Key Review Criteria Required for Acceptance?**

**Methods**

-Are the objectives of the study clearly articulated with a clear testable hypothesis stated?

-Is the study design appropriate to address the stated objectives?

-Is the population clearly described and appropriate for the hypothesis being tested?

-Is the sample size sufficient to ensure adequate power to address the hypothesis being tested?

-Were correct statistical analysis used to support conclusions?

-Are there concerns about ethical or regulatory requirements being met?

Reviewer #1: The authors responded to all my comments and made the necessary changes

Reviewer #2: (No Response)

Reviewer #3: (No Response)

**Results**

-Does the analysis presented match the analysis plan?

-Are the results clearly and completely presented?

-Are the figures (Tables, Images) of sufficient quality for clarity?

Reviewer #1: The authors responded to all my comments and made the necessary changes

Reviewer #2: (No Response)

Reviewer #3: (No Response)

**Conclusions**

-Are the conclusions supported by the data presented?

-Are the limitations of analysis clearly described?

-Do the authors discuss how these data can be helpful to advance our understanding of the topic under study?

-Is public health relevance addressed?

Reviewer #1: The authors responded to all my comments and made the necessary changes

Reviewer #2: (No Response)

Reviewer #3: (No Response)

**Editorial and Data Presentation Modifications?**

Reviewer #1: The authors responded to all my comments and made the necessary changes

Reviewer #2: (No Response)

Reviewer #3: (No Response)

**Summary and General Comments**

Reviewer #1: The authors responded to all my comments and made the necessary changes

Reviewer #2: (No Response)

Reviewer #3: There seem to have been a technical problem for my last comments to the first submission, so here they are again:

Figure 1: the additional records are 23 in the figure, but seems to be different from the number in S1 File. 

The diagnosis of taeniosis is very difficult. This should be emphasized more in the manuscript as it is a major limitation for the proper evaluation of drugs effectiveness. Include the methodology that was used for diagnosis of taeniosis in the figures/tables where possible. L592-593: which test would you suggest?

PLOS authors have the option to publish the peer review history of their article (what does this mean?). If published, this will include your full peer review and any attached files.

Reviewer #1: Yes: Ana Flisser

Reviewer #2: Yes: Daniel Alexis Zarate Rendon

Reviewer #3: No

---

## [Decision Letter · Decision Letter 2]

25 Oct 2019

Dear Dr. Haby,

We are pleased to inform you that your manuscript, "Systematic review of the effectiveness of selected drugs for preventive chemotherapy for Taenia solium taeniasis", has been editorially accepted for publication at PLOS Neglected Tropical Diseases.

Before your manuscript can be formally accepted and sent to production you will need to complete our formatting changes, which you will receive in a follow up email. Please note: your manuscript will not be scheduled for publication until you have made the required changes.

IMPORTANT NOTES

* Copyediting and Author Proofs: To ensure prompt publication, your manuscript will NOT be subject to detailed copyediting and you will NOT receive a typeset proof for review. The corresponding author will have one final opportunity to correct any errors when sent the requests mentioned above. Please review this version of your manuscript for any errors.

* If you or your institution will be preparing press materials for this manuscript, please inform our press team in advance at plosntds@plos.org. If you need to know your paper's publication date for media purposes, you must coordinate with our press team, and your manuscript will remain under a strict press embargo until the publication date and time. PLOS NTDs may choose to issue a press release for your article. If there is anything that the journal should know, please get in touch.

*Now that your manuscript has been provisionally accepted, please log into EM and update your profile. Go to http://www.editorialmanager.com/pntd, log in, and click on the "Update My Information" link at the top of the page. Please update your user information to ensure an efficient production and billing process.

*Note to LaTeX users only - Our staff will ask you to upload a TEX file in addition to the PDF before the paper can be sent to typesetting, so please carefully review our Latex Guidelines [http://www.plosntds.org/static/latexGuidelines.action] in the meantime.

Best regards,

Aysegul Taylan Ozkan, Ph.D., M.D.

Associate Editor

Jennifer Keiser

Deputy Editor

Reviewer's Responses to Questions

**Key Review Criteria Required for Acceptance?**

**Methods**

-Are the objectives of the study clearly articulated with a clear testable hypothesis stated?

-Is the study design appropriate to address the stated objectives?

-Is the population clearly described and appropriate for the hypothesis being tested?

-Is the sample size sufficient to ensure adequate power to address the hypothesis being tested?

-Were correct statistical analysis used to support conclusions?

-Are there concerns about ethical or regulatory requirements being met?

Reviewer #3: (No Response)

**Results**

-Does the analysis presented match the analysis plan?

-Are the results clearly and completely presented?

-Are the figures (Tables, Images) of sufficient quality for clarity?

Reviewer #3: (No Response)

**Conclusions**

-Are the conclusions supported by the data presented?

-Are the limitations of analysis clearly described?

-Do the authors discuss how these data can be helpful to advance our understanding of the topic under study?

-Is public health relevance addressed?

Reviewer #3: (No Response)

**Editorial and Data Presentation Modifications?**

Reviewer #3: (No Response)

**Summary and General Comments**

Reviewer #3: (No Response)

PLOS authors have the option to publish the peer review history of their article (what does this mean?). If published, this will include your full peer review and any attached files.

Reviewer #3: No

---

## [Editor Report · Acceptance letter]

6 Dec 2019

Dear Dr. Haby,

We are delighted to inform you that your manuscript, "Systematic review of the effectiveness of selected drugs for preventive chemotherapy for *Taenia solium* taeniasis," has been formally accepted for publication in PLOS Neglected Tropical Diseases.

Best regards,

Serap Aksoy

Editor-in-Chief

Shaden Kamhawi

Editor-in-Chief
